# Subsurface Insights of the Maricunga Gold Belt through Local Earthquake Tomography

Felipe Bugueño [1], Daniela Calle-Gardella [2,*], Diana Comte [2,3], Valentina Reyes-Wagner [2], Marcia Ojeda [2], Andreas Rietbrock [4] and Steven Roecker [5]

1 Departamento de Geología, Facultad Ciencias Físicas y Matemáticas, Universidad de Chile, Santiago 8370451, Chile
2 Advanced Mining Technology Center, Universidad de Chile, Santiago 8370451, Chile
3 Departamento de Geofísica, Facultad Ciencias Físicas y Matemáticas, Universidad de Chile, Santiago 8370449, Chile
4 Geophysical Institute, Department of Physics, Karlsruhe Institute of Technology, 76131 Karlsruhe, Germany
5 Earth and Environmental Sciences, School of Science, Rensselaer Polytechnic Institute, Troy, NY 12180, USA
* Correspondence: daniela.calle@amtc.cl

**Abstract:** With the advancement of the use of geophysical methods in mining exploration, the possibility of restudying known mineral deposits that could have greater potential than that previously estimated is opening up, as is the case in the Maricunga Belt (MB), which is a metallogenic belt located east of Copiapó, Chile, with a length of 200 km and oriented in the NNE-SSW direction. This belt hosts significant gold deposits classified as porphyry gold (-copper), epithermal gold (-silver) of a high sulphidation type, and transitional gold, in some districts. In this work we studied the characteristics of the MB through local earthquake tomography (LET), which revealed a clear spatial correlation between low Vp/Vs anomalies and the gold deposits, demonstrating that lithologic interpretation using Vp and Vs values of the seismic tomography makes sense for the most common rocks associated with the genesis of porphyry-type deposits. Furthermore, high Vp/Vs anomalies were correlated to the main regional faults around the study zone, which seem to have a robust structural control regarding the location of the deposits.

**Keywords:** Maricunga Belt; metallogenic belt; gold deposits; porphyry; mining exploration; local seismic tomography

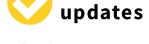



## 1. Introduction

Mining production in Chile is the top economic activity in terms of gross national product (producto interno bruto: PIB) [1], which generates an incentive to conduct further analysis on known mining deposits, as is the case for the Maricunga Metallogenic Belt (MB) in Chile. It is located east of Copiapó, with a length of 200 km oriented in the NNE-SSW direction. This belt hosts important gold deposits classified as epithermal gold (-silver), gold of high sulphidation type, and transitional and porphyry gold (-copper) in some districts [2], represented by yellow, orange, and red stars, respectively, in Figures 1 and 2.

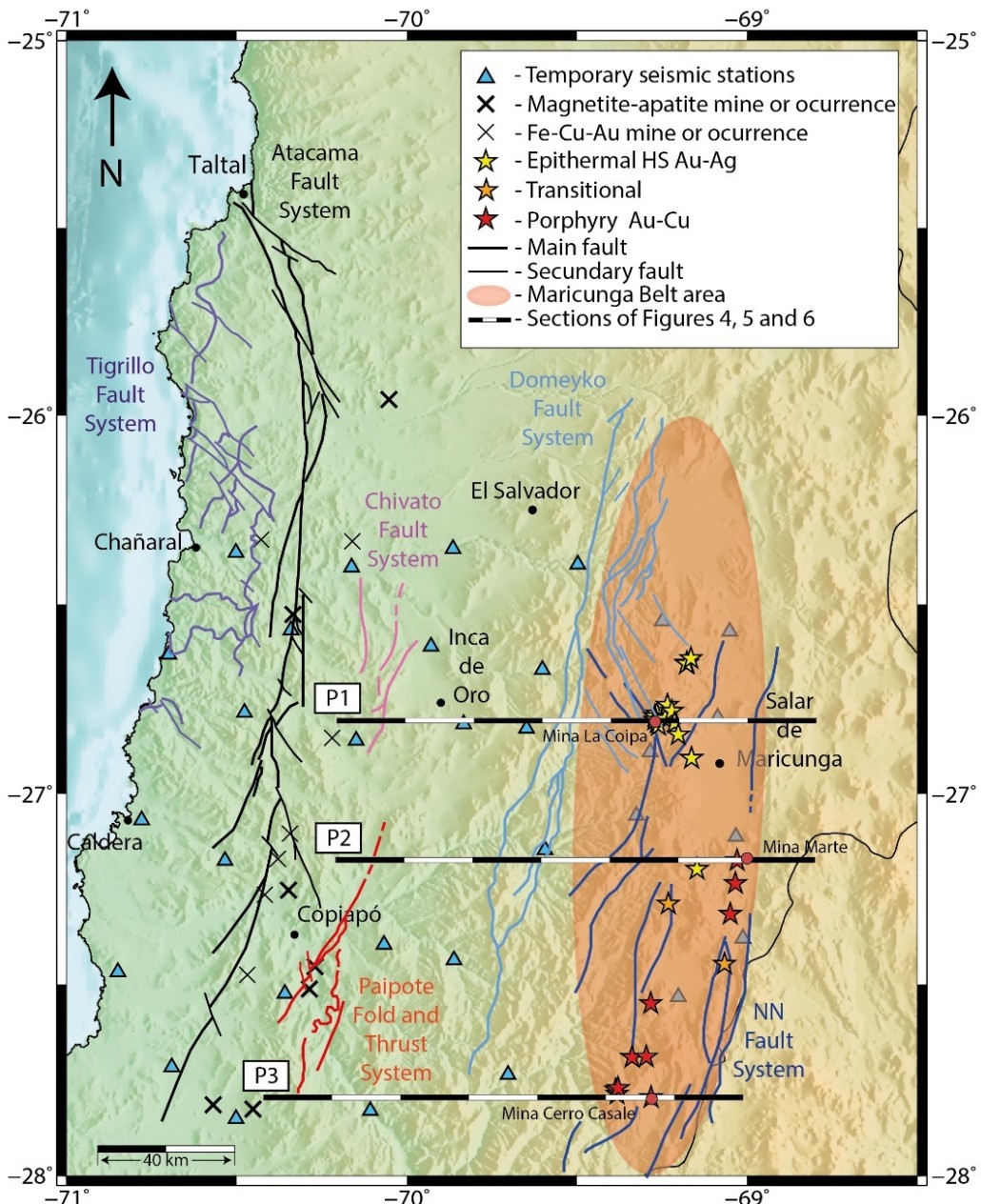

**Figure 1.** Overview map of the study area. The MB is represented by the red shadow area. Large regional mines and occurrences of magnetite-apatite and Fe oxide Cu-Au are indicated with light and dark crosses, respectively. Main regional-scale faults are indicated by purple (Tigrillo Fault System), black (Atacama Fault System), pink (Chivato Fault System), red (Paipote Fold and Thrust System), and light blue (Domeyko Fault System) lines (modified from [3]). The spatial distribution of gold deposits is indicated with yellow, orange, and red stars to represent high-sulphidation Au-Ag epithermal, transitional, and porphyry Au-Cu type deposits, respectively [2]. Selected local innominate (NN) fault systems are represented with blue lines (modified from [4]). Light-blue triangles indicate the temporary seismic network of the project developed between the research centers—the Advance Mining Technology Center (AMTC) of Chile and the Karlsruhe Institute of Technology of Germany in northern Chile [5,6]. Black-white lines outline the locations of the cross-sections through the 3-D velocity cube. Red dots refer to mines present in the same sections.

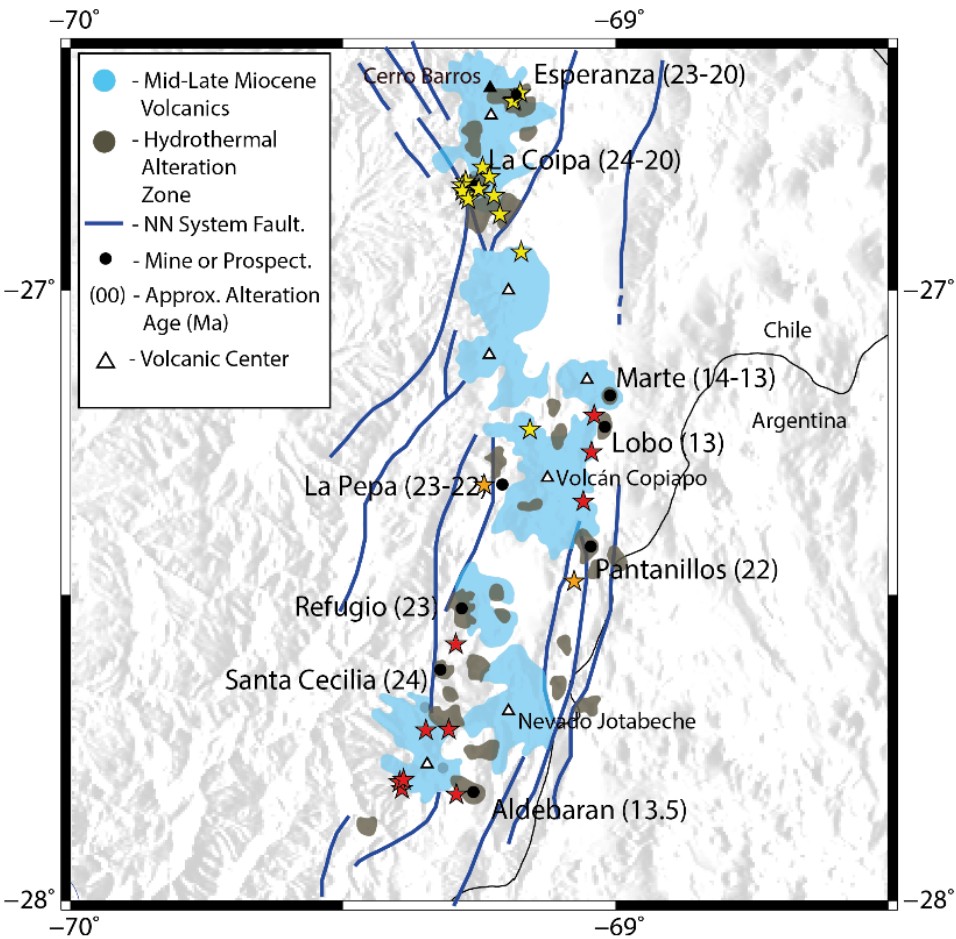

**Figure 2.** Location of alteration centers in the Maricunga Belt, northern Chile. The approximate ages of the main alteration-mineralization are shown as Ma. Hydrothermal alteration and Middle-Late Miocene volcanic zones are represented with gray and light blue shadows, respectively. Selected local innominate (NN) fault systems are represented with blue lines. Volcanic complexes are shown as white triangles. (Modified from [4]). The spatial distribution of gold deposits is indicated with yellow, orange, and red stars to represent high-sulphidation Au-Ag epithermal, transitional, and porphyry Au-Cu type deposits, respectively [2].

To better characterize the MB, in this study, we use local seismic tomography (LET), whose resulting velocity anomalies can be interpreted in terms of physicochemical property variations in the medium which has been applied, e.g., to mining exploration. Specifically, the Vp/Vs (P-wave and S-wave velocities ratio), which is independent of density and inversely proportional to rigidity, has been correlated with copper porphyries deposits in northern and central Chile. It has been observed that, for example, Chilean world-class porphyry-type deposits in Chuquicamata (northern Chile) and El Teniente (central Chile), and important Cu porphyries like those from Cerro Colorado (northern Chile) and Rio Blanco-Los Bronces (central Chile) correlate with low Vp/Vs anomalies [5].

## 2. Geotectonic and Metallogenic Context

The MB (red shadow area in Figure 1) corresponds to a transitional zone bounded on the north by a zone with normal subduction and the related Central Volcanic Zone (CVZ), and on the south by a zone with flat slab subduction. The transition of the subduction from the flat slab to a normal zone towards the north is relatively smooth while, towards the south, it is abrupt, according to the depths of the Wadati–Benioff zone [7].

The MB hosts mainly two mineralization types: epithermal gold (-silver) of high sulphidation type and porphyry-type gold (-copper), and their main characteristics are summarized below:

1.  High-sulphidation epithermal deposits (yellow stars in Figure 2) are located in the margins of volcanic complex Cerros Bravos (black triangle in Figure 2), exhibiting a dacitic to andesitic rock composition, and geochronological data indicate a mineralization age of about 23–18 Ma [4,8,9]. The mineralization is mainly structurally controlled by NE-SW oriented strike-slip faults (blue lines in Figure 2).

2.  Gold porphyry deposits (red stars in Figure 2) are located in the hillsides of the volcanic complexes Copiapó and Jotabeche (white triangles in Figure 2), having andesitic to dacitic rock composition, and their occurrences are characterized by two hydrothermal events (gray shadows in Figure 2). They are assigned to the late Oligocene–early Miocene (24–22 Ma) period, when La Pepa, Refugio, Santa Cecilia, and Pantanillo deposits were emplaced, and to the middle Miocene (14–12 Ma) period, when Cerro Casale, Lobo, and Marte deposits occurred [4].

In the study in [10], the genetic processes that lead to ore formation of porphyry copper-type deposits are recognized. In this context, the term "porphyry Cu" is used for any porphyry-type deposit related to one or more shallow intrusions that produce Cu as the principal commodity, i.e., including Mo- and/or Au-rich porphyry systems [11].

Arc magmas that are commonly found in spatial and temporal association with porphyry copper deposits are andesite, basaltic andesite, basalt, and potassium-rich magmas [11,12]. Primitive basaltic melts that are generated in the mantle wedge may ascend and pond at the mantle-crust boundary, where a combination of fractional crystallization, partial melting of lower crustal rocks, and extensive mixing between these magmas occurs [13], and andesites are produced. This is similar to the mixing, assimilation, storage, homogenization (MASH) model of [14] and is in accord with the presence of mafic and ultramafic cumulates at the crust-mantle boundary. At shallower depths, andesitic magmas may also be formed by physical mixing or mingling of felsic and mafic components in upper crustal magma chambers [15,16], which have been identified beneath almost every well-studied arc volcano [17], and appear to be an essential step in the formation of porphyry Cu deposits [10].

Along with effects of magmatic control on porphyry copper genesis [10] and hydrothermal control on metal distribution in porphyry Cu-(Mo-Au) systems [18], structural also control plays a crucial role in ore formation [19]. The latter includes: (1) the pre-, syn-, and post-mineral activation of existing structures locally, with varying senses of movement and development of new fracture patterns related to different events; (2) the creation of syn-mineral dilation to facilitate the flow of hydrothermal fluids containing entrained metals, usually to shallower crustal levels and cooler settings for mineral deposition; (3) the evolution of hydrothermal fluids to form different deposit styles at variable crustal settings; and (4) the effective mechanisms of metal precipitation.

According to [19], changes in the sensing of movement on geologic structures trigger the emplacement of intrusions (porphyry Cu-Au mineralization) and the evolution of rapidly rising hydrothermal fluids (high sulphidation epithermal fluids), locally with metals entrained in circulating groundwaters (low sulphidation epithermal fluids).

Based on selected geological structures from northern Chile, shown in Figure 1, spatial correlations between regional-scale fault systems (Atacama and Domeyko Fault systems, in black and light-blue lines in Figure 1, respectively) and (intrusion-related) magnetite-apatite and iron-oxide copper-gold deposits (light and dark crosses in Figure 1) are observed. It also shows that major faults structurally control the mineralization in the MB (Chivato Fault, and Paipote Fold and Thrust systems shown in pink and red lines in Figure 1, respectively).

## 3. Method: Local Earthquake Tomography (LET)

Seismic tomography data used in this work (Table S1 in Supplementary Material) are limited to an area between 26°30′ S to 28°00′ S, and between 68°30′ W to 70°30′ W and

belong to a larger seismic tomography dataset acquired within a joint project between two research centers—the Advance Mining Technology Center (AMTC) of Chile and the Karlsruhe Institute of Technology of Germany—presented in [20].

A brief summary of the project is given in the following. A seismic network of 88 short-period, three-component, continuous register stations was deployed in northern Chile (light blue triangles in Figure 1), and data were recording between 14 December 2018 and 17 January 2020. From the recorded seismological signals, a preliminary seismic catalog of P- and S-wave arrival times was determined by the Regressive ESTimator (REST) algorithm described in [21] using the 1D velocity model from [22]. The criteria used for this pre-catalog was a minimum of 4 detected phases and a maximum standard deviation of 1.5 s for all residuals of total travel times. From the complete original pre-catalog, 8152 possible events were localized inside our study area. Finally, by using the P- and S-wave arrival times of the initial catalog, 3D models of Vp, Vs, and Vp/Vs were created. For this purpose, first, the subsurface was parameterized in a 3D grid of 2 km wide nodes covering a volume of $295 \times 275 \times 150$ km$^3$ (W-E, N-S, and depth directions, respectively). Then, by using parameter events with a minimum of 30 recorded phases, a maximum of standard deviation of 0.5 s, and an azimuth of 200° as input, the joint inversion methodology described in [23,24] and applied by [21,25–27] was conducted, which consists of an iterative process, where hypocenter locations are first determined using an initial velocity model with the arrival times of the P-waves and S-waves, and then all hypocenters are jointly inverted to obtain an updated model. The final model was obtained after six iterations using a total of ~365.000 P-wave and ~323000 S-wave onsets from 12,930 events, having an RMS misfit for the final residuals of 0.1373 s.

## 4. Results

### 4.1. Vp/Vs

To analyze the behavior of the Vp/Vs in the studied area, depth slices through the model are presented at 2, 6, and 10 km, as shown in Figure 3.

At the 2 km depth (Figure 3a) two main low Vp/Vs anomalies can be observed. The Oligocene-Miocene gold deposits in MB are located around the center of the eastern anomaly. At greater depths (6 and 10 km depth, shown in Figures 3b and 3c, respectively), they merge into a single Vp/Vs anomaly. It is observed that the central part of the Vp/Vs anomaly moves to the west with increasing depth. This suggests that the epithermal deposits may be located further to the west at larger depths, which may indicate structural control by the major faults of the Domeyko Fault System.

Mineral deposits (stars in Figure 3) classified as high sulphidation gold-silver type (yellow stars in Figure 3) in the study area are mainly distributed in the northern part of the metallogenic belt, and are spatially associated with very low Vp/Vs (1.70–1.72). The porphyry gold and transitional types, distributed in the central and southern parts of the MB (red and orange stars, respectively, in Figure 3) are associated with a transitional zone toward higher values of Vp/Vs (1.73–1.76).

To deeply represent the MB cross sections of the Vp/Vs model are shown (Figures 4–6) crossing through the La Coipa (26°49′), Marte (27°10′), and Cerro Casale (27°47′) deposits, respectively (see black-white lines in Figure 1).

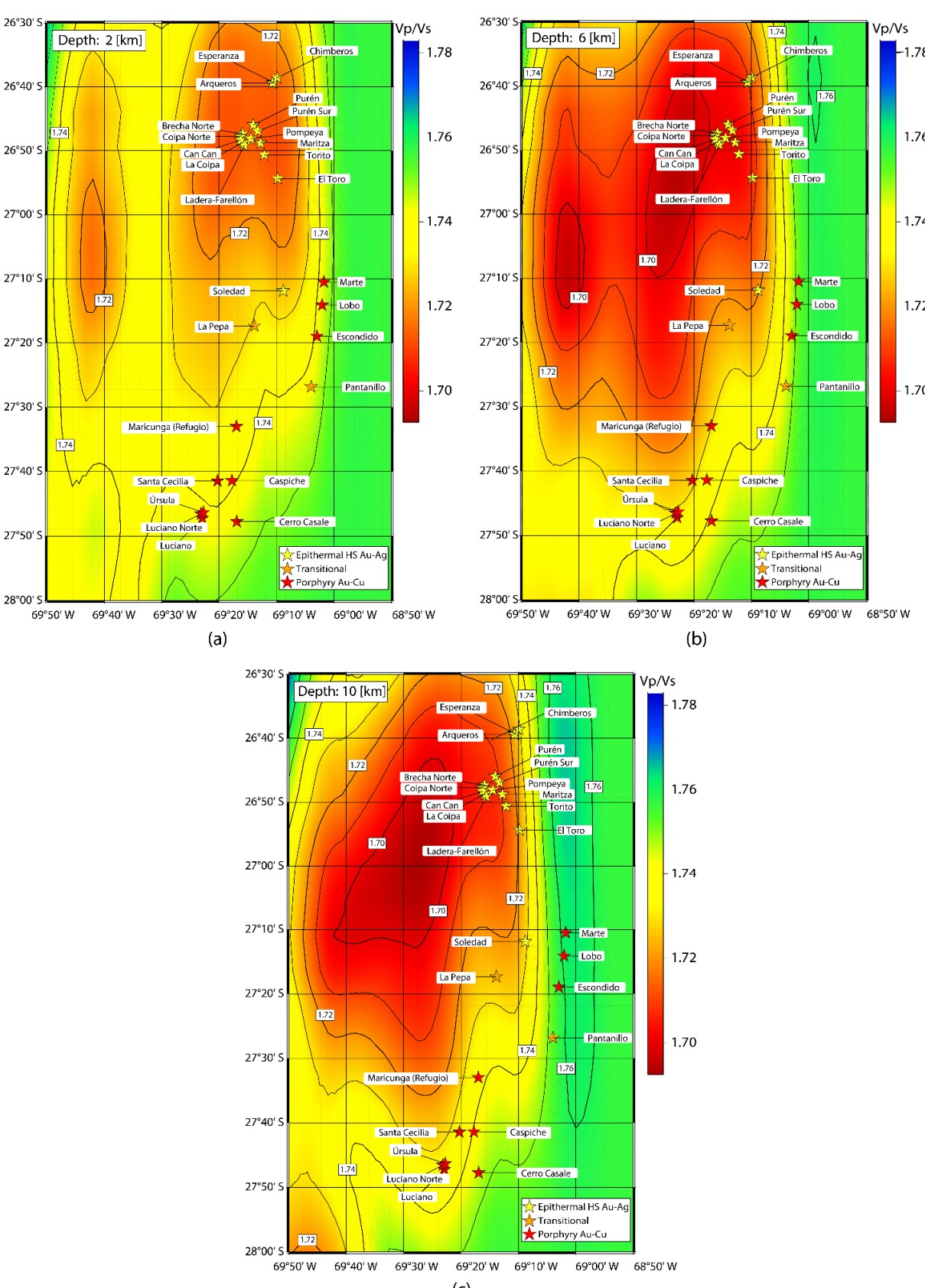

**Figure 3.** Slices through the Vp/Vs model (with contour intervals of 0.01), where (**a**) represents 2 km, (**b**) 6 km, and (**c**) 10 km depth. Yellow, orange, and red stars represent epithermal high-sulphidation Au-Ag, transitional, and porphyry Au deposits, respectively.

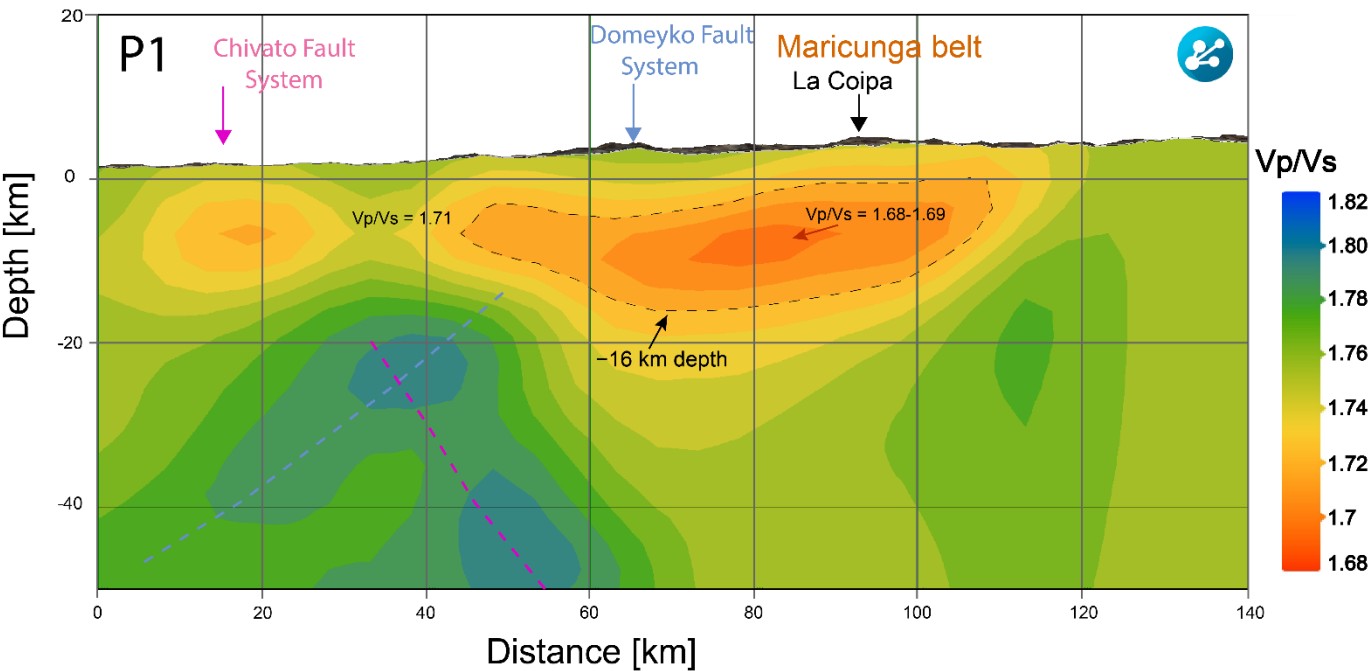

**Figure 4.** Cross section of the Vp/Vs model (with color intervals of 0.01) passing through the La Coipa deposit (26°49′) (see Figure 1). The locations of the Chivato and Domeyko Fault Systems are marked with pink and light blue arrows, respectively. The location of the La Coipa mine is marked with a black arrow. The location of the MB is marked with red letters. Light blue and pink dashed lines represent a possible deep projection of the Domeyko and Chivato Fault Systems, respectively.

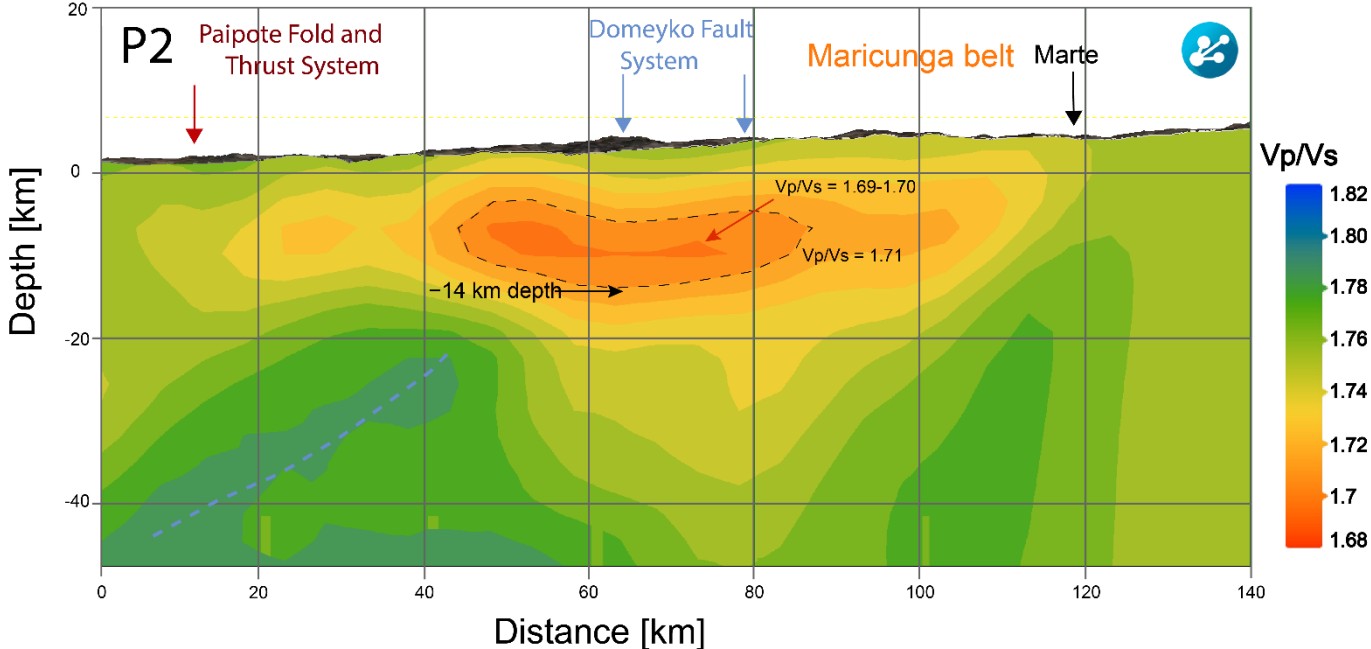

**Figure 5.** Cross section of the Vp/Vs model (with color intervals of 0.01) passing through the Marte deposit (27°10′) (see Figure 1). The locations of the Paipote Fold and Thrust System and the Domeyko Fault Systems are marked with red and light blue arrows, respectively. The location of the Marte mine is marked with a black arrow. The location of the MB is marked with red letters. Light blue dashed line represents a possible deep projection of the Domeyko Fault System.

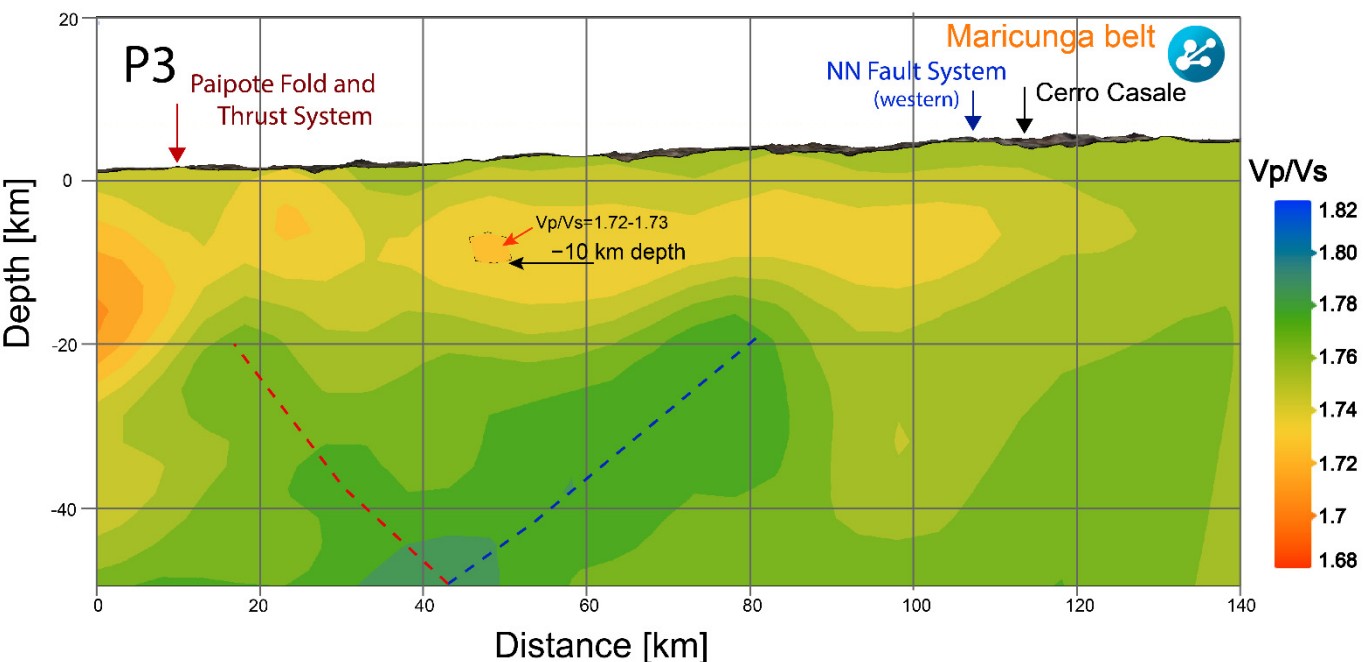

**Figure 6.** Cross section of the Vp/Vs model (with color intervals of 0.01) passing through the Cerro Casale deposit (27°47′) (see Figure 1). The location of the Cerro Casale mine is marked with a black arrow. The location of the MB is marked with red letters. Red and blue lines represent a possible deep projection of the Paipote Fold and Thrust System and the NN Fault System, respectively.

As observed, there is a progressive weakening of the low Vp/Vs anomaly from north to south of the MB, while the anomaly also decreases in depth.

Additionally, some of the main regional faults in the study area can be correlated to high Vp/Vs at larger depths of about >20 km, such as at the Domeyko Fault System and the Chivato Fault System (light blue and pink lines, respectively in Figure 1), which are clearly visible in sections that cross the La Coipa (26°49′) and Marte (27°10′) deposits (light blue and pink dashed lines in Figures 4 and 5), which also seem to have strong structural control on the low Vp/Vs anomaly at shallower depths. At the southern part of the belt, the cross section of Cerro Casale (27°47′) (Figure 6) shows a less intense low Vp/Vs anomaly, and the high Vp/Vs may be correlated to the Paipote Fold and Thrust System (red dashed lines in Figure 6), which controls some Cretacic deposits, such as Candelaria. In addition, intermediate Vp/Vs values with an east direction could be associated with the NN Fault System (blue dashed lines in Figure 6).

### 4.2. Vp and Vs

The determined Vp and Vs models can be used to interpret the lithologies for all deposits located in the belt, following the work of Christensen [28].

For this purpose, we use the Vp and Vs experimental measurements for a variety of igneous, metamorphic, and sedimentary rocks distributed worldwide, under a 200 MPa pressure, collected from Christensen [28] (blue dots in Figure 7). These Vp and Vs are represented by average velocities, according to rock type. In addition, we considered the observed Vp and Vs values of the gold deposits of MB (extracted from the final velocity models) at a 6.5 km depth, which approximately simulates the pressure used by [28] (orange dots in Figure 7). Finally, we used Brocher's regression empirical formula from [29]:

$$V_{S\_Brocher}\left(\frac{km}{s}\right) = 0.7858 - 1.2344V_p + 0.7949V_p{}^2 - 0.1238V_p{}^3 + 0.0064V_p{}^4 \quad (1)$$

to infer Vs as a function of observed Vp (gray dots in Figure 7). The comparative analysis (based on Table 1) is shown in Figure 7, where a good correlation between Vs values derived from Brocher's formula with observed Vs values can be seen.

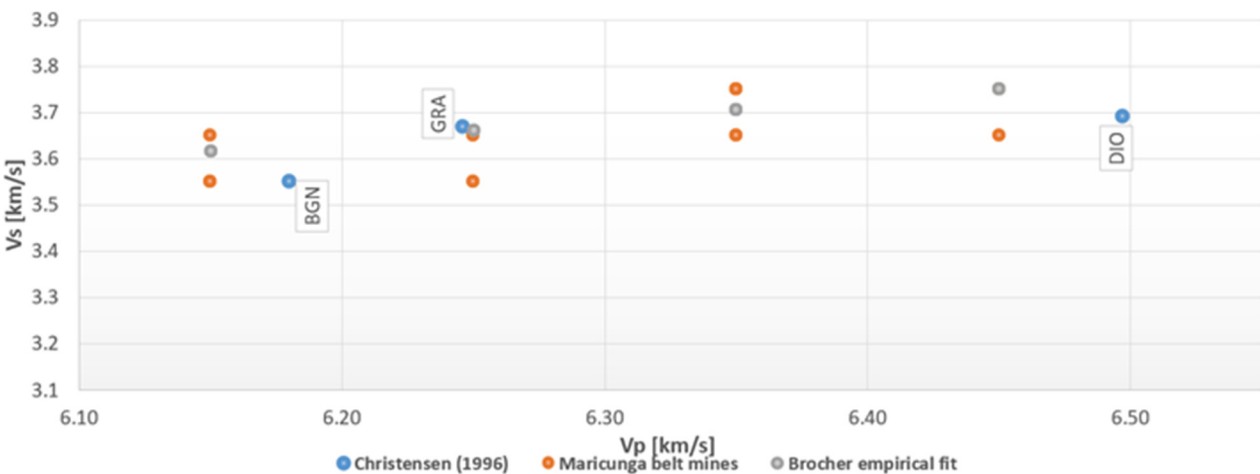

**Figure 7.** Vs is plotted as a function of Vp for biotite (tonalite), gneiss (BGN), diorite (DIO), and granite-granodiorite (GRA) lithologies. Blue dots represent the average Vp and Vs values obtained experimentally by Christensen [28] under a pressure of 200 MPa. Orange dots represent the Vp and Vs values observed in the MB gold deposits obtained from the final velocity models at a 6.5 km depth. The grey dots represent the ratio between the observed Vp and the Vs fitted according to the formula from Brocher (Equation (1)).

**Table 1.** Information regarding the main lithologies and seismic velocities of the main gold deposits in the MB, including deposit type, age, Vp, Vs, and Vp/Vs values obtained from the original velocity model at 6.5 km depth, Vs velocities derived from the formula from Brocher [29] calculated from Equation (1), and the lithology associated with the interpretation of these velocity values.

| Mine | LON [°] | LAT [°] | Deposit Type | AGE [Ma.] | This Work | | | | |
|---|---|---|---|---|---|---|---|---|---|
| | | | | | Vp/Vs | Vp | Vs | Vs_Brocher | Lithology |
| Arqueros | −69.178 | −26.657 | Epithermal HS Au-Ag | 23.2–19.3 | 1.715 | 6.150 | 3.650 | 3.618 | BGN |
| Brecha Norte | −69.265 | −26.789 | Epithermal HS Au-Ag | 17.3 | 1.705 | 6.350 | 3.750 | 3.707 | GRA-DIO |
| Can Can | −69.271 | −26.805 | Epithermal HS Au-Ag | | 1.705 | 6.450 | 3.750 | 3.751 | DIO |
| Caspiche | −69.296 | −27.690 | Porphyry Au-Cu | 25.4 | 1.735 | 6.450 | 3.650 | 3.751 | DIO |
| Cerro Casale | −69.282 | −27.796 | Porphyry Au-Cu | 13.9 | 1.745 | 6.450 | 3.650 | 3.751 | DIO |
| Chimberos | −69.165 | −26.643 | Epithermal HS Au-Ag | | 1.715 | 6.150 | 3.550 | 3.618 | BGN |
| Coipa Norte | −69.271 | −26.796 | Epithermal HS Au-Ag | | 1.705 | 6.350 | 3.750 | 3.707 | GRA-DIO |
| El Toro | −69.163 | −26.906 | Epithermal HS Au-Ag | | 1.705 | 6.150 | 3.550 | 3.618 | BGN |
| Escondido | −69.049 | −27.316 | Porphyry Au-Cu | | 1.745 | 6.250 | 3.550 | 3.663 | GRA |
| Esperanza | −69.176 | −26.655 | Epithermal HS Au-Ag | 23–20 | 1.715 | 6.150 | 3.650 | 3.618 | BGN |
| La Coipa | −69.264 | −26.812 | Epithermal HS Au-Ag | 24.7 | 1.705 | 6.350 | 3.750 | 3.707 | GRA-DIO |
| La Pepa | −69.231 | −27.289 | Transitional | 23–22 | 1.725 | 6.350 | 3.650 | 3.707 | GRA-DIO |
| Ladera-Farellón | −69.260 | −26.819 | Epithermal HS Au-Ag | 20.2 | 1.705 | 6.350 | 3.750 | 3.707 | GRA-DIO |
| Lobo | −69.034 | −27.235 | Porphyry Au-Cu | 13 | 1.755 | 6.150 | 3.550 | 3.618 | BGN |
| Luciano | −69.382 | −27.787 | Porphyry Au-Cu | | 1.735 | 6.350 | 3.650 | 3.707 | GRA-DIO |
| Luciano Norte | −69.386 | −27.776 | Porphyry Au-Cu | | 1.735 | 6.350 | 3.650 | 3.707 | GRA-DIO |
| Maricunga (Refugio) | −69.283 | −27.550 | Porphyry Au-Cu | 23 | 1.735 | 6.450 | 3.750 | 3.751 | DIO |
| Maritza | −69.213 | −26.813 | Epithermal HS Au-Ag | | 1.705 | 6.250 | 3.650 | 3.663 | GRA |
| Marte | −69.029 | −27.175 | Porphyry Au-Cu | 14–13 | 1.755 | 6.25 | 3.550 | 3.663 | GRA |
| Pantanillo | −69.066 | −27.447 | Transitional | 22 | 1.745 | 6.25 | 3.550 | 3.663 | GRA |
| Pompeya | −69.241 | −26.802 | Epithermal HS Au-Ag | | 1.705 | 6.35 | 3.750 | 3.707 | GRA-DIO |
| Purén | −69.234 | −26.766 | Epithermal HS Au-Ag | | 1.705 | 6.35 | 3.750 | 3.707 | GRA-DIO |
| Purén Sur | −69.221 | −26.782 | Epithermal HS Au-Ag | | 1.705 | 6.25 | 3.650 | 3.663 | GRA |
| Santa Cecilia | −69.338 | −27.691 | Porphyry Au-Cu | 24 | 1.735 | 6.45 | 3.750 | 3.751 | DIO |
| Soledad | −69.147 | −27.198 | Epithermal HS Au-Ag | | 1.725 | 6.15 | 3.550 | 3.618 | BGN |
| Torito | −69.202 | −26.844 | Epithermal HS Au-Ag | | 1.705 | 6.25 | 3.650 | 3.663 | GRA |
| Úrsula | −69.379 | −27.772 | Porphyry Au-Cu | | 1.735 | 6.35 | 3.650 | 3.707 | GRA-DIO |

Table 1 shows a summary of the lithologies interpreted for all deposits located in the MB, along with the Vp and Vs values derived from the models and obtained by regression. The Vp values are in the range of 6.1–6.4 km/s for deposits located in the northern and central parts of the belt, which are in agreement with the results in [28], and along with low Vp/Vs values, could represent intrusive rocks of granite-granodiorite composition. Towards the south of the belt, the Vp values are in the range of 6.4–6.5 km/s, which could represent an intermediate rock composition, such as dioritic rock.

Predicting lithologies based on Vp and Vs modeled values is not the aim of this research; however, the results presented in this work suggest a very good correlation between Vp/Vs anomalies, their Vp and Vs related values, and the intermediate to felsic intrusive suite rocks studied by Christensen [28]. In fact, in [7], porphyries and volcanic products are well documented for the main deposits in the MB—andesite, diorite, dacite, ryodacite, and rhyolite—all of them from the intermediate to felsic suite. There is a trend which shows that Vp is slightly higher in the south of the MB (Table 1), which may reflect a change in rock composition.

It should be emphasized that experimental measures on some lithologies in [28], such as basaltic facies and meta-sedimentary rocks, have not been taken into account due to the fact that all Vp/Vs values obtained in the models are too low (1.70–1.75) to be correlated with these lithologies (1.76–1.86). This is also supported by the processes that occur in the earth's mantle and crust. In fact, the tectonic frame in which the study area is located corresponds to an active continental margin and magmatic processes such as MASH (melting, assimilation, storage, homogenization) and AFC (assimilation-fractional crystallization) that allow primitive magmas to evolve when they rise up to the earth's crust. Thus, magmas achieve intermediate (diorite) to felsic (tonalite, granodiorite, granite) compositions which are documented in the study area.

## 5. Discussion and Conclusions

In terms of the Vp/Vs, this study shows a high correlation with the location of the mineral deposits that comprise the Maricunga Metallogenic Belt (MB). High-sulphidation gold-silver epithermal deposits, such as those found in the La Coipa and Esperanza districts in the northern part of the belt, are located just above the center of low Vp/Vs anomaly, with values around 1.70–1.72. On the other hand, porphyry gold and transitional deposits, such as those found in the Marte-Lobo, Escondido, La Pepa, Pantanillo, Refugio, Cerro Casale, and the cluster Úrsula-Luciano-Luciano Norte in the central and southern part of the belt, are located in the transitions to higher Vp/Vs, with values around 1.73–1.76.

The center of a great low Vp/Vs anomaly is observed at 10 km depth, which is divided into two anomalies at shallower levels. This may provide some information about the genetic similarities between mineral deposits that comprise the MB and potentially, older mineral deposits that could be discovered around the western anomaly.

In deep sections of La Coipa, Marte, and Cerro Casale, the high Vp/Vs anomalies indicate poorly consolidated or highly fractured rocks that can be correlated to some regional faults having an important structural control on emplacements of intrusive rocks and mineralizing fluids. The Domeyko Fault System can be associated with cross sections passing through the La Coipa (26°49′) and Marte (27°10′) deposits, with a west verging high Vp/Vs anomaly, which seems to be controlled by this fault system. Some local faults which control mineralization may also be correlated to high Vp/Vs anomalies.

Along with Christensen's [28] results, which show Vp and Vs measurements for a wide variety of lithologies, it is possible to make a correlation between these data and the those determined in this work. According to this correlation, gold deposits located in the northern and central parts of the belt are associated with felsic to intermediate intrusive rocks (tonalite, granodiorite, and granite), while deposits located in the south of the belt are associated with intermediate rocks (diorite). This segmentation along the belt may indicate that the volcanic arc at the northern and central parts of the belt experienced conditions in such a way that magmas interacted more with crustal rocks than did those located at

the southern part of the belt. Even though Vp and Vs values at regional scales do not predict lithologies, all plotted samples seem to be correlated with a more evolved felsic to intermediate composition, which makes sense with the genesis of gold-rich porphyries and high-sulphidation epithermal deposits that comprise the MB. Furthermore, it shows a very good correlation with volcanic and intrusive rocks found and documented around the mineral deposits.

In summary, seismic tomography for the MB shows a clear spatial correlation between low Vp/Vs anomalies and the types of gold deposits found there. Lithologic interpretation using Vp and Vs values makes sense in regards to the most common rocks associated with the genesis of porphyry-type deposits; and high Vp/Vs anomalies were correlated with to the main regional faults around the study zone, which seem to provide a robust structural control of the emplacement of the deposits.

**Supplementary Materials:** The following supporting information can be downloaded at: https://www.mdpi.com/article/10.3390/min12111437/s1, Table S1: Final velocity model used on this paper.

**Author Contributions:** Conceptualization, D.C.; methodology, D.C. and D.C.-G.; software, S.R.; validation, D.C. and F.B.; formal analysis, F.B.; investigation, F.B., D.C., D.C.-G. and M.O.; resources, A.R. and D.C.; data curation: V.R.-W.; writing—original draft preparation, F.B.; writing—review and editing, D.C.-G. and D.C.; visualization, F.B. and D.C.-G.; supervision, D.C.-G. and D.C.; project administration, D.C.; funding acquisition, D.C. All authors have read and agreed to the published version of the manuscript.

**Funding:** This research was funded by the National Agency for Research and Development of Chile (ANID) by Project AFB180004, Project AFB220002 and by the FONDEF ID21I10022 project.

**Data Availability Statement:** The data presented in this study are available on request from the corresponding author.

**Acknowledgments:** We would like to acknowledge Andrea Navarro, who leads the work from which this paper is extracted, and who contributed to the final revision of this paper. We thank the team that provided the installation, maintenance, and retirement of the temporary seismological network used for this work, with special thanks to Sergio León-Ríos, Francisco Pastén-Araya, Gerardo Peña, and Alejandro Faúndez.

**Conflicts of Interest:** The authors declare no conflict of interest.

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
