# Peer review of "Subsurface Insights of the Maricunga Gold Belt through Local Earthquake Tomography"

_minerals, doi:10.3390/min12111437_

Round 1
Reviewer 1 Report
Dear authors,
I thank you for the interesting read, your paper is clear and easy to read, however I have a slight concern that I feel should be adressed in your paper.
The results from LET seem of course and reliable, however I'm concerned about how you associate different minerals/deposite type with such slight differences in Vp and Vs. How can you be sure that your models are resolved enough and that the uncertainty associated with these velocity updates permit you to do such a correlation. I underline the fact that sometimes you are associated different deposits based on a change of 0.1 km/s which at this scale could be well within your margin of error (Wheter the data are fitted or not, depending on the ill-posedness of your LET governed by the acquisition etc.).
I would appreciate that you at least point out this issue for the readers,
Best regards.
Author Response
To Reviewers,
We want to thank you for the time spent reading the initial draft and the suggestions provided. Most of the writing and spelling suggestions that appear in an annotated version of the original manuscript were taken into account and changed for this new version. The suggestions for improving images were also incorporated.
Regarding the issues discussed in the paper, both Reviewersagreed that there was a piece of missing information on how the tomography was obtained. I think it was not evident in the first draft. Still, the tomography used in this paper is an extract or part of a regional tomography that is part of another study that will be exposed at AGU 2002 (Navarro-Aránguiz A., et al. Seismic Tomography in the Chilean Pampean Flat-Slab Segment: Latitudinal Differences in the Double Seismic Zone and their Relationship with Low Coupling Zones. AGU Fall Meeting, Chicago, USA, Dec. 2022; American Geophysical Union (AGU) 2022), and it will be published. It is for this reason, and out of respect for my colleague, that I cannot go into more detail on the tomography, however, together we expanded the description of the methodology used for a better understanding of the reader.
And about the query of the Reviewer 1 about the section on how the lithology is related to Vp and Vs velocity model, we want to clarify that the suggested correlation was made based on theoretical studies of rocks (explained at greater in this updated draft) and that they can be related to the lithology already known in the area. However, it Is important to emphasize that it is a secondary analysis to the main topic of this paper which is the correlation of the Vp/Vs with the already-known mineral deposits.
Once again we thank you and remain attentive to possible new comments.
Daniela CalleA
AMTC Researcher
Universidad de Chile
Reviewer 2 Report
I find the topic and results interesting, most of the article focused on the geological and mineralogical background, but I would have liked to see more geophysics. I have two ask two things from the authors, besides the comments already found in the PDF:
1) Expand the "Method" section. It is only one paragraph mentioning the method(s) used. It should include more about why you used the method, any implementation decisions or parameters used and why, and the geophysics behind it.
2) I strongly suggest you get the whole text re-read by a native English speaker, some sentences and expressions seem direct translation from Spanish and do not make much sense when read in English.

Author Response
To Reviewers,
We want to thank you for the time spent reading the initial draft and the suggestions provided. Most of the writing and spelling suggestions that appear in an annotated version of the original manuscript were taken into account and changed for this new version. The suggestions for improving images were also incorporated.
Regarding the issues discussed in the paper, both Reviewersagreed that there was a piece of missing information on how the tomography was obtained. I think it was not evident in the first draft. Still, the tomography used in this paper is an extract or part of a regional tomography that is part of another study that will be exposed at AGU 2002 (Navarro-Aránguiz A., et al. Seismic Tomography in the Chilean Pampean Flat-Slab Segment: Latitudinal Differences in the Double Seismic Zone and their Relationship with Low Coupling Zones. AGU Fall Meeting, Chicago, USA, Dec. 2022; American Geophysical Union (AGU) 2022), and it will be published. It is for this reason, and out of respect for my colleague, that I cannot go into more detail on the tomography, however, together we expanded the description of the methodology used for a better understanding of the reader.
And about the query of the Reviewer 1 about the section on how the lithology is related to Vp and Vs velocity model, we want to clarify that the suggested correlation was made based on theoretical studies of rocks (explained at greater in this updated draft) and that they can be related to the lithology already known in the area. However, it Is important to emphasize that it is a secondary analysis to the main topic of this paper which is the correlation of the Vp/Vs with the already-known mineral deposits.
Once again we thank you and remain attentive to possible new comments.
Daniela Calle
AMTC Researcher
Universidad de Chile